# CITYZER Observation Network and Data Delivery System

Walter Schmidt[1], Ari-Matti Harri[1], Timo Nousiainen[1], Harri Hohti[1], Lasse Johansson[1], Olli Ojanperä[2], Erkki Viitala[3], Jarkko Niemi[4], Jani Turpeinen[5], Erkka Saukko[6], Topi Rönkkö[7], Pekka Lahti[8]

[1] Finnish Meteorological Institute, Erik Palménin aukio 1, 00560 Helsinki, Finland
[2] Vaisala Ltd, Vanha Nurmijärventie 21, 01670 Vantaa, Finland
[3] Emtele Ltd, Tampere, Finland
[4] HSY Helsinki, Ilmalantori 1, 00240 Helsinki, Finland
[5] Sasken Finland Oy, Vissavedentie 1, 69600 Kaustinen, Finland
[6] Pegasor Ltd, Hatanpään valtatie 34 C, 33100 Tampere, Finland
[7] Tampere University of Technology, Kalevantie 4, 33100 Tampere, Finland
[8] Haaga-Helia University of Applied Sciences, Ratapihantie 13, 00520 Helsinki, Finland

*Correspondence to*: Timo Nousiainen (timo.nousiainen@fmi.fi)

**Keywords: system concept, sensor networks, weather forecast, air quality, mobile applications**

**Abstract.** CITYZER develops new digital services and products to support decision making processes related to weather and air quality in cities. This includes, e.g., early warnings and forecasts (0-24 h), which allow for avoiding weather-related accidents, mitigate human distress and costs from weather-related damage and bad air quality, and generally improve the resilience and safety of the society. The project takes advantage of the latest scientific know-how and directly exploits the expertise obtained from earlier projects. Central to the project is the Observation Network Manager NM10 developed by Vaisala, on which CITYZER defines and builds new commercial services and connects new sensor networks, e.g. for air quality measurements, as well as the ENFUSER local scale air quality modelling system developed by the Finnish Meteorological Institute, for real-time air quality forecasts and nowcasts.

## 1 The CITYZER ecosystem concept

### 1.1 Background

The atmosphere plays a central role in the global circulation of heat, water and volatiles, making our current life possible and convenient. It is the reservoir of the breathable gases, and takes away the exhaled gases that would otherwise poison our immediate surroundings. Our civilization is closely linked with the atmosphere in a multitude of ways, whether we are considering simple agricultural communities or modern urban areas.

The CITYZER project concentrates on two closely linked aspects of the atmosphere with the focus on urban climate. On the one hand, we consider air quality, which is highly important in particular due to its possible adverse health effects. On the other hand, we consider weather phenomena, in particular those linked with precipitating weather systems. Weather events

can cause episodes of high pollutant concentrations in urban areas, or they can help clean the urban air through washing and wet deposition or by bringing in cleaner air from adjacent areas. In their own right, the weather effects can be hazardous, causing infrastructural damages and loss of lives, or they may be bothersome and cause harmful friction for logistics planning and the everyday life of human society. Collecting in near real time sensor data from the area of interest, sophisticated interpolation and forecast software provides reliable information for the next hours concerning air quality and weather phenomena, which can be made available to the public or service providers via mobile application or warning systems (Harri et al., 2018), (Smart & Clean, 2020).

## 1.2 CITYZER ecosystem

The main objective of the project was to develop new digital services and products to support decision making processes related to weather and air quality. The overall idea was to create a new ecosystem of services open to third-party developers, based on open data. To that end, the project set out to design and implement an IoT-based platform for collecting, refining and delivering environmental data. Accordingly, the project was closely linked to several megatrends, namely Open data, Big data, Platform economy and Digitalization.

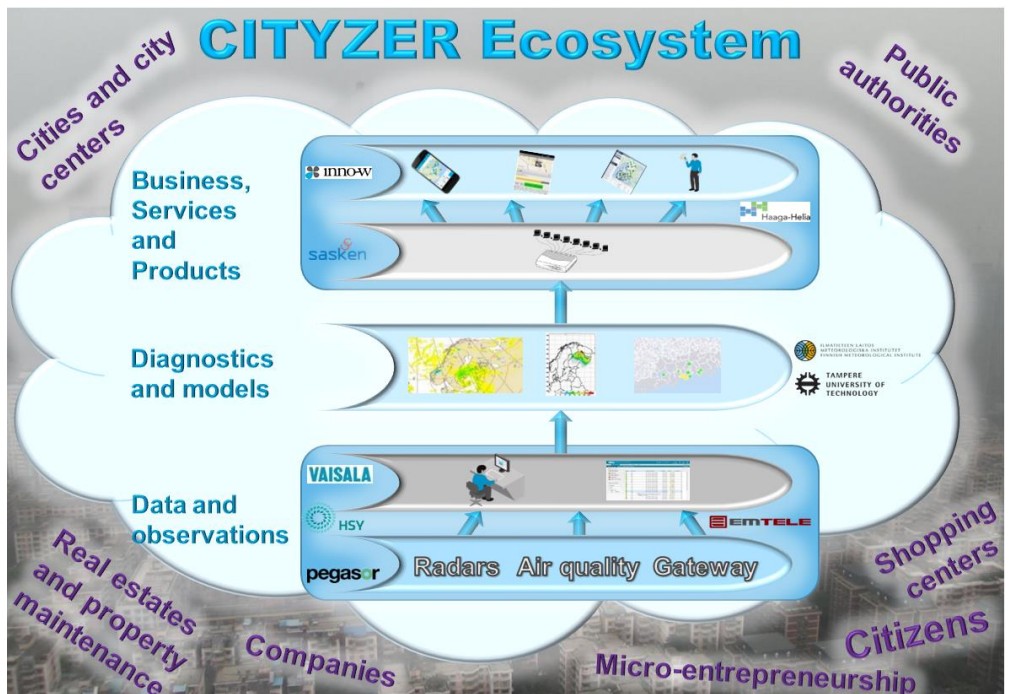

**Figure 1: CITYZER Ecosystem – the mentioned contributors are part of the CITYZER consortium**

The CITYZER ecosystem consists of three levels:

(1) Measurement data and observation level containing the data collection and management systems like sensor networks and public or restricted data repositories

(2) Diagnostics and modelling level where data are collected, analyzed and prepared for a user friendly utilization

(3) Services and product levels which utilizes subsets of the raw or refined data, possibly in combination with external data sources like map service data, to present information in a form and update frequency suitable for the user like authorities or the public

This article concentrates on the first and the third level leaving the details of the analysis forecast software for weather and
air quality to other publications.

## 2. CITYZER system architecture

### 2.1 General structure

The system is very modular with well-defined interfaces allowing the replacement, removal or addition of different elements
and their implementation in different environments from commercial hardware configurations to virtual cloud-based networks.

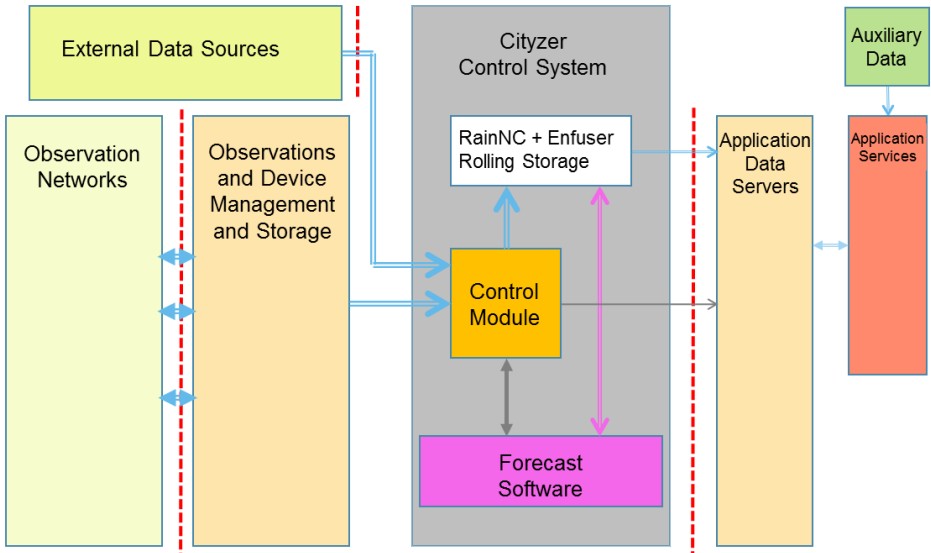

**Figure 2: System Architecture, Module View**

The six main modules of the CITYZER system architecture are presented in Fig. 2.

(1) Observation Networks with various sensors for all observation parameters to be included into the specific deployment together with the field data networks for forwarding the measurement data and controlling the sensors from remote supervising stations where applicable

(2) Observations and Device Management and Storage with automatic or manual sensor supervision, data collection and possibly data re-formatting services

(3) CITYZER Control System with a data storage covering about 1 week of sensor and modelling data, the control module which fetches new data once available and controls the forecasting software.

(4) Access to External Data Sources like public data providers for weather-related data

(5) One or several Application Data Servers providing a standardized interface between the stored data and external application providers

(6) One or several application service providers which allow users to access the information in a user friendly way from stationary or mobile devices. The service providers might additionally access other auxiliary data sources like map services to render the data in a user-optimized way.

## 2.2 Forecast software

Depending on the needs or the deployment one or several forecast model software packages can be implemented. These might reside inside the same physical computer system as the control system or could be implemented on different platforms and linked to the control system via file sharing technology. In the course of the project the precipitation nowcasting software for rainfall was installed and operated on the same Linux-based server as the control module. It was adapted from the modelling software developed at the Finnish Meteorological Institute as part of the RAVAKE project (Heinonen et al., 2013). The source code size was 205 kByte with a size of the executable file of 180 kByte. A typical run time for one complete precipitation nowcast covering the larger Helsinki area is about 8 to 13 sec. A motion vector analysis for the complete area of Finland takes about 1 minute.

The air quality forecast modelling software ENFUSER (ENFUSER, 2020), (Johansson, 2015) was developed into an operational modelling system with test implementations for the CITYZER demonstrator in a virtual Linux environment, as cloud service and on a local Windows computer. It includes detailed treatment of traffic emissions for individual roads, shipping emissions and elevated point sources such as power plants taking into account urban morphology, atmospheric stability and rain nowcast information.

## 2.3 Data interfaces

Sensor data can be provided in two different ways:

(1) Public data accessible via standard web interfaces. The CITYZER environment implements the Open Geospatial Consortium's Web Feature Service (OGC / WFS) as described in the organisation's document OGC 04-094, (OGC, 2020) reference list. Most European and many world-wide weather and air quality service data are available using this standard. Access lists are published among others by the EU (EEA public map services, 2018). Data are also available via commercial services providing among others processed hazard monitoring data from space-born platforms like the SENTINEL satellites or Landsat.

(2) Dedicated sensor networks covering the area of interest. These might be owned by different authorities and private providers and will usually need specialized hardware interfaces and protocol translation software before they can be integrated into the CITYZER data storage module. Various solutions as deployed in the Helsinki area are described below. They are based on a network manager either integrated with a group of sensors or provided separately as a network controller, which then uses the same OGC protocol as the public data for feeding the real-time data into the data storage module. A similar approach is also used for other application areas, like the sensor network for agriculture and water monitoring in Southern Finland (Kotamäki, Niina et.al., 2009).

The sensor data are collected in the data storage in a unified format and sorted according to data type as grid-data or coordinate-based point data, day of observation and sensor providing the data.

The analysis and forecast software modules are ingesting the data needed for the forecast interval adapting to the extended or reduced availability of sensors automatically. The forecast results are written back into the same data storage using a format compatible to the sensor data, but organized into different files for easy access.

A database system keeps track of the available data and provides this information on request to user applications. These may be mobile applications allowing the public to assess the weather and air quality situation in the near future in an area of interest. The data related to the inquiry are then provided as links to the respective file in the data storage for download by the application server. This usually will also access geographic data from external providers as platform on which the weather and air quality data are presented to the end user.

An alternative access to the forecast data was implemented using the open source SmartMet Server interface (SmartMet, 2020) developed at the Finnish Meteorological Institute in 2013 as implementation of the INSPIRE (INSPIRE, 2020) requirements for open data access. This interface was used already by several applications outside the framework of the project.

## 2.4 System security considerations

One critical aspect of any multi-interface system is the system and data security. Most of the sensor systems deployed in the field can usually be compromised without excessive efforts. The same is true for interfaces between service providers and end-user application systems like mobile phones. The CITYZER system architecture therefore provides hardware solutions

for protecting the central data storage and forecast software resources. Any connection for incoming sensor data or requests from application servers is separated by a strict firewall without any external write access possibility. The control system is sending information requests to each connected data provider at regular intervals. When new data are available it fetches these data, filters them through incoming filter software and writes the possibly re-formatted data into its internal data storage, thereby eliminating any illegal data or possible command instructions.

The availability of new sensor or model output data is indicated to the application database outside the firewall, providing a complete link to these data. All data storage files are accessible by the application database software via a hardware read-only link without any write-back possibility. In case of a compromised database it can be re-built on-the-fly from the data inside the secure data storage.

Access to the application data server is controlled via an authentication mechanism. Each possible user has to register first with this server indicating also the type of access intended: data polling and fetching by the application or automatic push service according to rules to be defined by the user service like update time, geographical boundaries parameters to be provided etc.

## 2.5 Data Flow Control

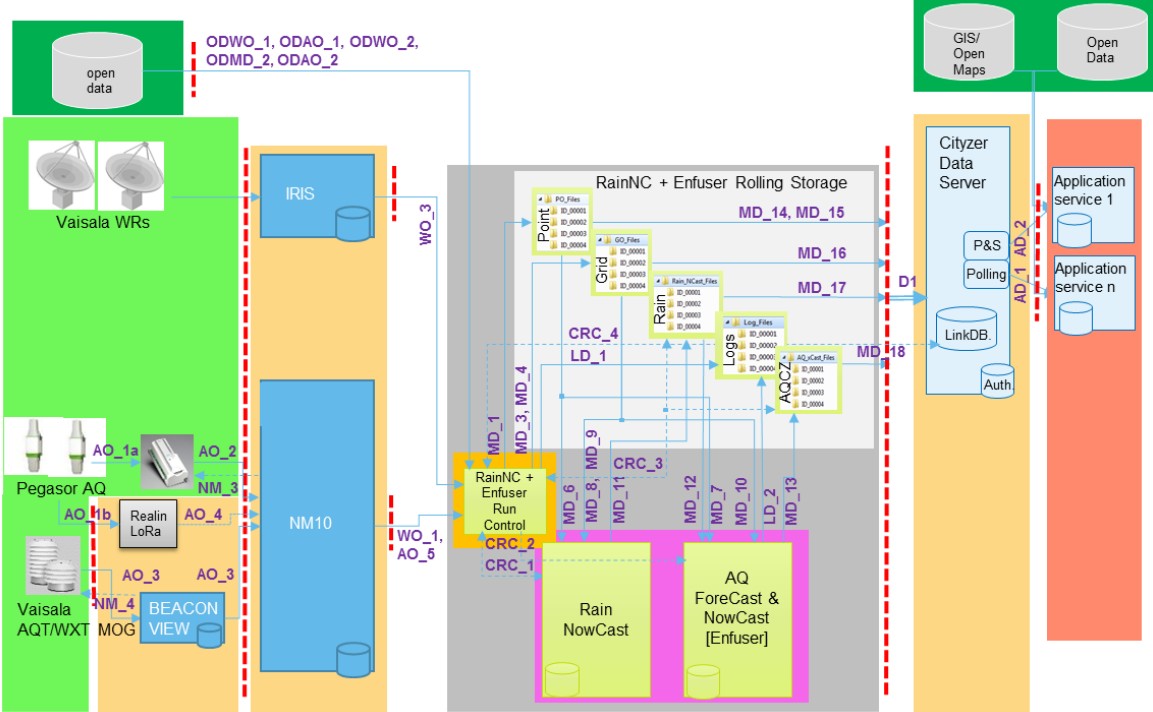

**Figure 3: CITYZER Demo Deployment Architecture**


In the example diagram, Fig. 3, one set of air quality sensors from Pegasor is connected via a public phone data network or via an Emtele-provided LoRa (LoRa Alliance, 2020) sub-network to Vaisala Observation Network Manager NM10. Another set of air quality sensors, AQT400 series from Vaisala, is connected via a Vaisala Beacon View data access and control system to the NM10. Weather radar data are collected separately via the common Vaisala IRIS system. The connection

between NM10 and the second layer control system is realized as Open Geospatial Consortiums' Web Feature Service (OGC/WFS) to make it as universally adaptable as possible. Also other public data servers are connected to the control system via OGC/WFS compatible communication standards. The abbreviations used in the flow diagram are OD for open data, AO for air quality observations, WO for Weather observations, MD for model data and CRC for command flow from the Control Module.


The Control Module synchronizes all activities in the CITYZER system. Based on regular timing or action events different processes are started, data are collected or the database updated thereby indicating to the application processes that new data are available.

The initial configuration is defined as follows – see the timing information in the leftmost column in Fig. 4.

Once a minute requests are sent via OGC protocol to the attached and registered data providers to check the availability of new data. In case of new data these are fetched, converted if necessary and stored in the respective file system of the server according to type, sensor and time of observation. All incoming data are expected to be calibrated in standard physical units. The data sets usually contain also quality information indicating if the related sensors were correctly calibrated or off-line. Such an update is also indicated to the database system to make observational data directly available to the users, possibly

initiating an alert to the application servers if so requested.

Every five minutes new weather radar data are fetched, stored in the file system and the rain nowcasting model is activated. Its results are available inside a minute and stored in a different part of the file system. This information is also forwarded to the database system. At a spatial resolution of 250 m the size of one radar composite file varies between 50 and 1000 kByte depending on the amount and type of observed precipitation. A forecast output  motion vector field covering the whole area

of Finland has a typical size of 60 kByte.

Every 60 minutes the Air Quality data modelling system is activated providing new hourly and one day forecasts of the air quality development in the covered area. These forecasts make use both of fresh observational data as newly calculated rain nowcast results anticipating air quality modifications by imminent rainfall. One typical model run for the Helsinki area takes about 35 minutes and generates maps with a grid size of 13 m x 13 m. These approximately 3 million grid cells contain

concentration estimates for $NO_2$ and $O_3$, dust concentrations Pm10 and Pm2.5 and Air Quality Index (AQI) estimates. For each forecast about 1000 local air quality measurements are ingested and used besides the hourly 10 km resolution weather forecast maps and the 100 m resolution / 5 min time resolution rain nowcasting data.

Alternatively any analysis software package might run autonomously at certain time intervals accessing those data available

at the start of a new activation.

The output of each control script is logged. Figure 4 shows the various data flow paths where the solid arrows indicate the flow of data between the different modules, the dashed lines represent command flows for synchronization of the different modules and control of the attached sensors where possible and needed.

In the demonstration implementation the central control module is implemented as a set of scripts activated according to

specific timing rules.

A typical control and time diagram is shown in Fig. 4.

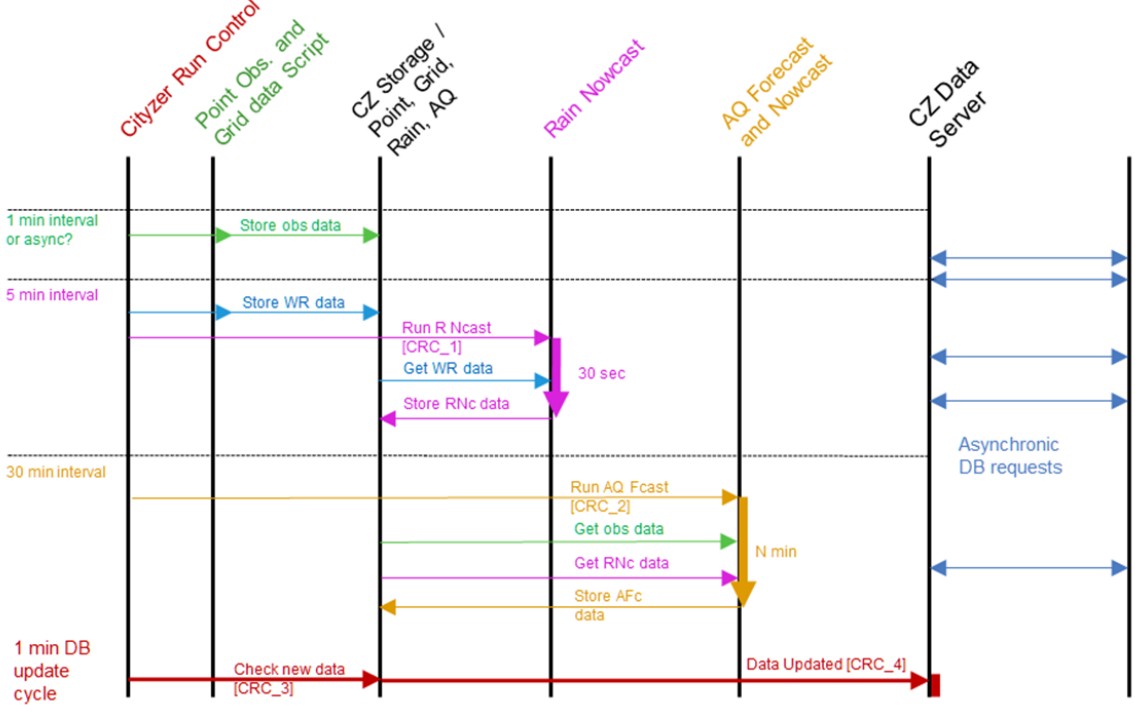

**Figure 4: CITYZER Data and Command Flow Control**

The combination of different data sources as input to model predictions requires that the provided data are collected in the

same time frame compatible with the time resolution of the forecast models. With rain data nowcasts updated roughly once every 5 minutes and air quality forecasts once every 30 minutes all used data sources should use a time base consistent with these constraints. Other implementations with different types of observed parameters like water flow or lighting occurrences might need different precisions of time synchronisation. This could be achieved by locally implemented GPS-based, radio link or internet-provided time information. The error detection part of the CITYZER control system should exclude data with

time information deviations incompatible with the needs of the used modelling software.

## 2.6 User Application Access

User applications register usually with an application service provider which provides a downloadable application for displaying the requested data and a service which generates these data according to the requirements of the end user. The application server has to register with the CITYZER data base to gain access to the data. For the demonstrator a public-domain Linux version of the MySQL data base system is used, but also other data base systems are possible. At the time of registration the strategy of accessing new data can be defined either as notification to get alerted whenever data relevant for the application are updated or as polling, where the application requests data matching certain criteria, and the data base provides these data if available. In both cases the returned information provides a complete data link address from where a file with the related data can be retrieved. Typical repetition times for polling or automatic notification generation is in the order of minutes, but can be adjusted by the user or application service provider to reduce the data flow or increase the near-real-time information. If a notification trigger level is defined by the user, an alert is generated inside a minute after the condition is met. Also this parameter is freely adjustable when setting up the system.

The application access scheme is shown in the Fig. 5.

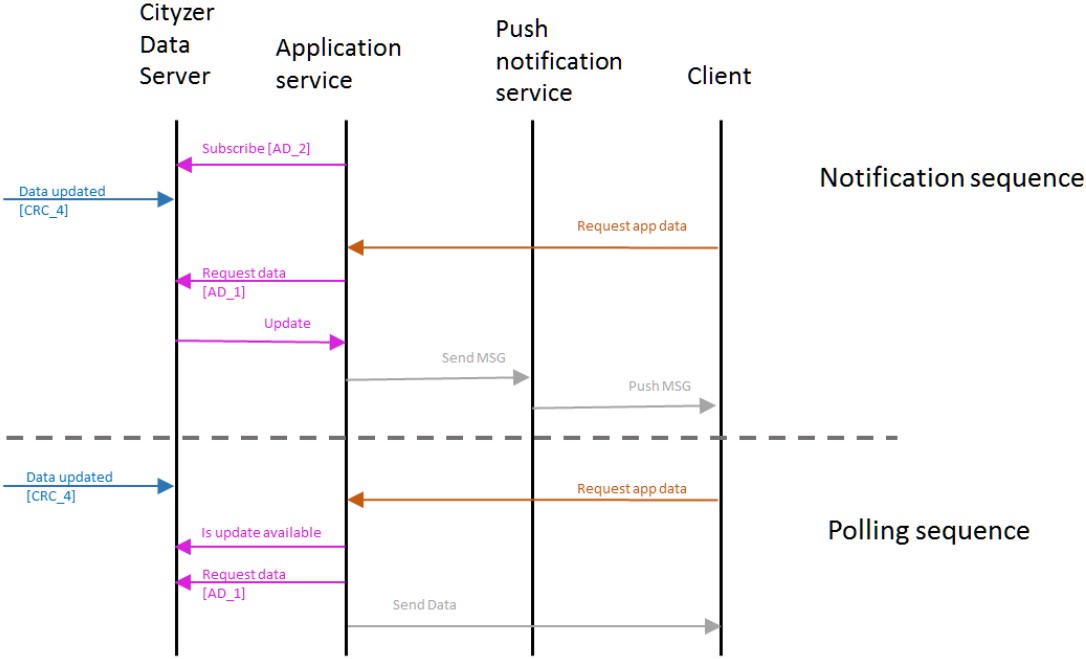

**Figure 5: CITYZER User Application Access Scheme**

The system is designed to support also the generation of automatic alerts. If one or several parameters exceed pre-defined limits an alert message can be generated and sent to the connected mobile phone or user service center. This feature is currently implemented in the Helsinki area to re-schedule the street cleaning services in case of microdust levels exceeding pre-defined levels on major streets before they may become a health hazard.

**2.7 Data Storage Structure**

The rolling storage of data is divided into two separate filesystems:

- A local storage for incoming and cache-type data on a local disk. Storage size of the demo version: 8 GByte.

- Shared storage for output data in an external filesystem, which is mounted writable for internal use, and read-only for the
CITYZER database server. The shared storage directory is physically mounted on an externally visible mount point. Its absolute path must be used by the application server software when sending notifications of new data to the CITYZER database server in Java Script Object Notation format (JSON, 2020). Storage size of the demo version: 120 GByte.

Radar data for rain nowcasting are stored in the ODIM HDF5 format (ODIM, 2020), developed by the European Meteorological Service Network EUMETNET. Data originating from the various air quality sensors and air quality model
data for single geographic points are stored in the XML-based GML format (GML, 2020), also endorsed by the INSPIRE directive. Co-ordinate grid data are stored in NetCDF format (NetCDF, 2020).

In the local storage there is a directory "Today" which is linked every day at midnight to the actual directory of the new day (Mon, Tue, Wed etc.). Each daily directory is structured as follows:

| Nowcasting | Contains data for rain nowcasting model Ravake |
|---|---|
| Sub-directories containing data in ODIM HDF5 format: | |
| /Rain | Incoming radar reflectivity composite files |
| /Vectors | Incoming motion vector fields of radar composites |
| /Probability | Output rain accumulation exceedance probability fields |
| | |
| /Ravake | Output rain intensity fields from rain nowcasting |
| | Files *.dat contain ensemble member data and are used only by the exceedance probability analyzer |
| | Files *.h5 are in ODIM HDF5 format and contain deterministic rain rate nowcast data. |
| ENFUSER | Contains model grid and point observation data for ENFUSER AQ model input |
| /grid_data | |
| /HIRLAM | Incoming NWP model HIRLAM data in NetCDF |
| /SILAM | Incoming AQ model SILAM data in NetCDF format |
| /Probability | Output rain accumulation exceedance probability fields |
| /point_data | |
| /FMI-AQ | Incoming AQ observation data from FMI stations in GML format |
| /NM10 | Incoming AQ observation data from Vaisala NM10 in GML format |

The Linux crontab-based data fetch process of the Control Module copies and links new files from these directories for notification and fetching for CITYZER database server. The fetch process checks the status of dedicated log files in the shared output directories at defined time intervals (currently 10 minutes) to copy new data files to storage and link them for notification.

## 3. CITYZER Ecosystem Implementations

### 3.1 CITYZER Demonstration Implementation

As verification of the CITYZER ecosystem concept all components were implemented in the Helsinki area (see also Fig. 3). Existing air quality sensor networks were augmented by additional sensors and their various data connections. Real-time data from the Finnish weather radar network were used as input for the rain forecast software. The data server together with the modelling software and the application database system were built using resources of the Finnish Meteorological

institute. A dedicated application for web access and mobile phones was developed showing the forecast information for rain and air quality parameters projected onto a map of the larger Helsinki area and allowing also the specification of automatic alerts in case a specified parameter is going to exceed a pre-set limit. In preparation for the deployment of the demonstration system several air quality measurement campaigns were executed, e.g. (Hietikko et al., 2018), (Järvinen et al., 2019), (Kuula et al., 2019), (Teinilä et al., 2019). An example of an air quality forecast map for the larger Helsinki area is shown in Fig. 6.

The corresponding rain nowcast map for the same area can be seen in Fig. 7, while the application was configured to show the probability for light rain. Alternatives for medium and heavy rain are also offered. An additional in-built service allows the end user to receive an alert via mobile phone when the probability for rain at a user-specified point will exceed a given level.

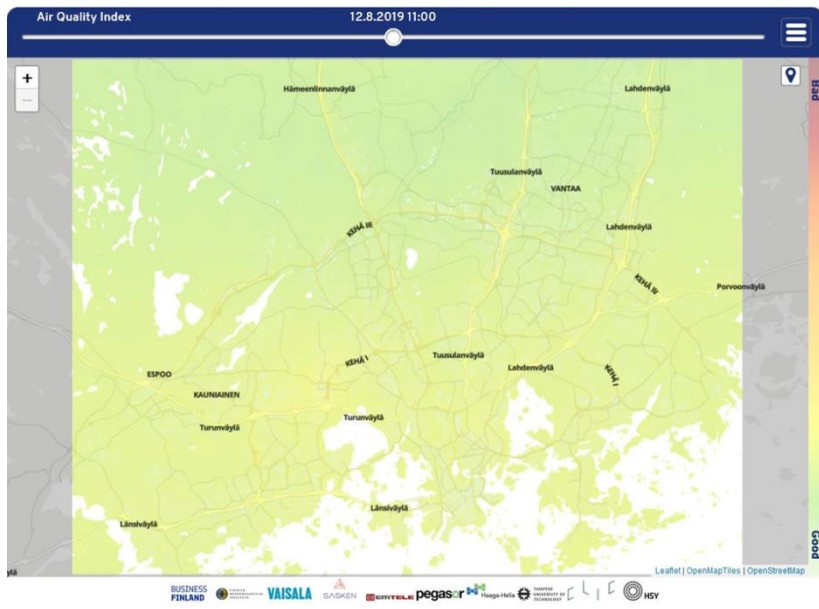

Figure 6: Air Quality Index forecast for the Helsinki area

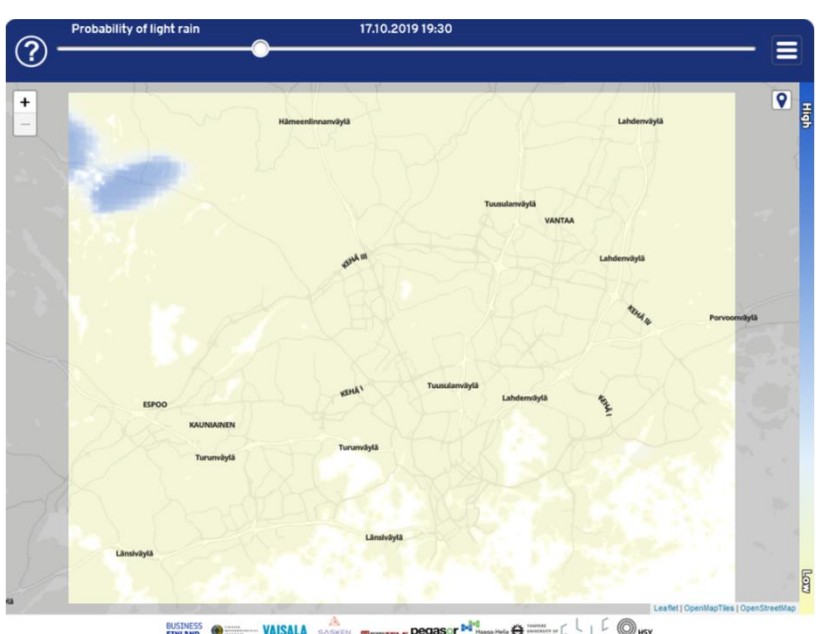

Figure 7: Rain nowcast for the Helsinki area

### 3.2 HAQT (HELSINKI metropolitan Air Quality Testbed)

A subset of the CITYZER ecosystem concentrating on the forecasting of air quality parameters was developed to an operative service, implemented and maintained by the Finnish Meteorological Institute. It started its routine operation in the beginning of 2019 and is among other applications used by the City of Helsinki to optimize the deployment of street maintenance and cleaning resources for keeping the dust level along traffic routes at a low level (HAQT, 2020). Updated air quality forecasts for the Helsinki region are also shown routinely on the electronic advertisement panels in the Helsinki metro trains (HSY Air quality map, 2020).

### 3.3 NAQT (Nanjing AQ Testbed)

A modified version of the HAQT environment was developed and deployed in the Nanjing region of China, see (Harri et al., 2018)

### 3.4 HOPE (Healthy Outdoor Premises for Everyone)

The main purpose of the EU-funded project HOPE is to empower citizens to develop their own districts. The project focuses on three different districts with varying air quality challenges: One district with heavy port related traffic, another with street canyons and with over 40 000 daily vehicles, and the third district which is affected by main roads and wood burning at homes. Change that the project wants to achieve is that the citizens will find air quality issues easily relatable and understandable by creating a feedback loop between high-resolution hyperlocal air quality data and actions of individuals and communities (HOPE, 2020).

### 3.5 Haaga-Helia Student-generated Applications

During the CITYZER project, students of the Haaga-Helia University of Applied Sciences in Helsinki were asked to develop mobile applications for air quality and rain forecast monitoring based on the CITYZER demonstrator implementation. Using an in-house developed learning ecosystem for business information technology the students successfully demonstrated that new service concepts and mobile applications can be easily developed and linked to the data services provided by the CITYZER environment.

### 3.6 Scientific utilization of the CITYZER Ecosystem

Any implementation of the CITYZER Ecosystem can be used as a versatile tool to support scientific research in a wide range of environmental fields. With the simultaneous availability of detailed measurement data covering one week and forecasts based on these data the quality of the used forecast models can be checked and improved and the influence of many observable and assumed parameters can be analyzed. The capability of the implemented forecast models to provide interpolated spatial data across the covered area extends the range of available environmental monitoring stations and e.g.

allows the detailed correlation between pollution sources and their consequences for the air quality in the region or the effects of geographical and urban features on the spatial development of heavy rain situations. The modularity of the system supports also the concurrent usage of alternative forecast models to find the best approach for a given application.

Using the error detection possibilities based either on hardware-provided failure information or model-based probability estimates, problems in the data collection system can be detected at an early stage and prevented from affecting the provided forecast results and related misinterpretations of weather and environmental phenomena.

### 3.6 Conclusions

The CITYZER project set out to develop a new ecosystem of products and services related to weather and air quality in cities. Central to this objective was the CITYZER platform, which would enable these services.

In the project the overarching system architecture was designed such that the platform would be scalable and modular thereby allowing the easy inclusion of other environmental parameters, addition and removal of sensor networks and optimization of applications for a wide range of potential users. In addition, data security was given much attention. Based on this system architecture a deployment architecture was then designed for the demonstration implementation. It included all the key modules so that the overall end-to-end functionality of the platform could be tested and demonstrated, but did not include all the peripherals and optional functionalities. As an example, lightning observations and their forecasts were left out from the deployment version.

Integrating a wide range of sensor systems with a central server suitable for different geopolitical areas faces the challenge of interface incompatibilities. The CITYZER platform solved this problem by implementing as server interfaces only a small set of international data format standards, which can be easily generated by sensor network controllers. System and data integrity was ensured by preventing any active outside access into the central server and strictly filtering any incoming data against possible illegal contents. With this approach similar systems can be established without major effort in different parts of the world and for different purposes supporting the growing demands for environment related problem management.

To prove the flexibility of the system the demonstrator was implemented such that different modules were physically located at different places. For example, the NM10 network manager was running in a Vaisala-operated infrastructure, while the prediction models and data storages were located at FMI and the user interface in the Amazon cloud.

The modularity of the architecture was tested with the HAQT and NAQT projects, which both employed partial implementations of the platform (only air quality part). The NAQT project also tested the transferability of the platform, as a new local implementation of the CITYZER control system in Nanjing was necessary due to data transfer restrictions from China.

Overall, it can be concluded that the CITYZER platform, both the architecture and the implementation, met the requirements set on it. It also attracted additional funding, such as the HAQT and NAQT projects, where it could be further tested and developed. Currently, the platform is undergoing some changes as some modules are being moved into operationally

supported infrastructures. This will be necessary especially for attracting third-party services to the CITYZER ecosystem or integrating it into developing Smart City (Smart Cities – Smart Living, 2020), (Smart City Helsinki, 2018) concepts.

## Author contributions

Schmidt, Walter: Project administration, methodology, visualization, writing – original draft

Harri, Ari-Matti: Funding acquisition, project administration, supervision, conceptualization

Nousiainen, Timo: Project administration, supervision, writing - review & editing

Hohti, Harri: Data curation, resources, software, writing – original draft section

Johansson, Lasse: Methology, data curation, software, writing – original draft section

Ojanperä, Olli: Project administration, methodology, resources, visualization

Viitala, Erkki: Resources

Niemi, Jarkko: Resources

Turpeinen, Jani: Resources, data curation, software, writing – original draft section

Saukko, Erkka: Resources

Rönkkö, Topi: Methology, resources

Lahti, Pekka: Resources, writing – original draft section

## Competing interests

Authors Ari-Matti Harri and Walter Schmidt are members of the editorial board of the GI-journal as associate editors. The other authors declare that they have no conflict of interest.

## Acknowlegment

The CITYZER research project was funded by Business Finland (Business Finland. 2020), Dnro 3021/31/2015, and contributions of the project consortium partners Finnish Meteorological Institute (FMI, 2020), Tampere University (Tampere University, 2020), Haaga-Helia University of Applied Sciences (Haaga-Helia, 2020), Vaisala Oyj (Vaisala, 2020), Pegasor Oy (Pegasor, 2020), Sasken Finland Oy (Sasken, 2020), Emtele Oy (Emtele, 2020) and Helsinki Region Environmental Services Authority (HSY, 2020).

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
