# Peer review of "CITYZER Observation Network and Data Delivery System"

_Geoscientific Instrumentation, Methods and Data Systems, 2020_

## Referee Comment (RC1) · Anonymous Referee #1 · 14 May 2020

The weather effects can be hazardous and require close monitoring and forecasting. It is even more important in the urban areas with highly concentrated population. For these areas not only the weather but air quality is a very important factor. CITYZER Observation Network and Data Delivery System designed and implemented by authors provides early warnings and short-term forecasts to weather and air quality which allow to avoid weather-caused accidents and generally improve the health of the society. I think the system is well designed, takes advantage of the existing automated network management software which significantly reduces amount of resources necessary for developing and implementation. The modularity with well-defined interfaces for and between elements that allows implementation in different hardware and software configurations is another strong feature of this system. It is very good to see that authors

dedicated a lot of attention to the security aspect. Unfortunately, a lot of the field sensors and IoT devices can be relatively easy compromised and jeopardize the whole system. Implementation of the firewall without external write access possibilities looks like an adequate solution. It is very exciting that a dedicated web and mobile applications to access the forecast information for rain and air quality was developed and used in verification of the CITYZER ecosystem concept. Ability of an end user to receive alerts via mobile phone when the probability for rain at a user-specified point will exceed a given level is a well thought feature of that application. This is a very interesting and well written paper that should be of wide interest. The authors should be congratulated on their accomplishments in designing and successfully implementing this system.
* * *

---

## Referee Comment (RC2) · Anonymous Referee #2 · 30 May 2020

**1   Discussion**

This paper introduce a technical presentation of the CITYZER Observation Network and Data Delivery System. CITYZER is a project funded by multiple governmental and private organizations.

It aims to provide refined and raw weather data to end user (as corporation or population) in order to enhance weather hazard policies.

The overall quality of the paper is good and the high level process description is done. As for all technical studies it's a bit frustrating to have only high level details (focused on the core system and not on the sensor side or the data), but in this case the whole

mechanism can be apprehended well. Most of all, various implementation are presented, which can show the claimed modularity of the system and its uses day by day which is clearly an interesting point.

One of the greatest achievement of the paper is the end-to-end system of notification and polling which show the usefulness of the web service paradigm in such scientifics fields, especially the OGC WS*. The use of generic container as HDF and NetCDF for the data storage is also a good message.

Another contribution could be a scientific point of view au the usefulness of such system in addition to the commercial and governmental point of view. But maybe this is dedicated to another publication as the text mention it.

This paper is then accepted with **minor revision which could improve the overall paper**. Congrats to the authors.

**2 Revisions**

**Introduction**:

The aim of the paper is well introduced but the state of the art could be enhanced with a little bit more papers about sensors systems (such as systems used in researches in kriging methods for land survey, or maximum entropy for sensor positioning, OGC based systems,, SmartCity sensing systems or even SENTINEL hazard monitoring commercial and public systems for example)

**Section 1.2CITYZER ecosystem:**

Fig1 : The figure 1 appears pixelated and it can be hard to link the text description and the figure elements. This comment can be done for all figures. It's not easy to link description and figures elements (mismatch in naming etc) and they are often blurred.

**Section 2**

Which database are used ? PostGIS ? Does the interfaces (even if WFS is used) are in some-ways affected by any Vaisala data format ?

**Section 2,5 Data flow**

Authors have shown a great use of WFS web service. A mention of point to point sensors using SOS et auto declaration of sensors thanks to SPS (used or not) could be an upgrade.

Around the line 162 authors mention the synchronization of the system, it can be interesting to know what time base use the system. For example, is the platform use an unified time base acquired through GPS system? Or all the modules and sensors have their own time base? What about the models time base ?

Time monitoring in such application is a useful information to some future end users.

On the figure 5 the polling sequence appears to flow counter clockwise. Is it inverted ? If not can the frequency of isAviableData() and RequestData() functions be precised ?

**Conclusions:**

Even if the paper is accepted with minor revision a decent conclusion (achievement, main goal, difficulties, usages and scientific/technical contributions to the state of the art etc... ) have to be included.

---

## Author Comment (AC1) · 30 Jun 2020

The authors thank the reviewers for their positive and very helpful comments. Below are detailed answers to the comments and suggestions of the reviewers.

While the main emphasis of the project was the development of an operational environment for dealing with environmental hazards, the implemented concept can also be exploited for several theoretical research purposes at it allows both the comparison of the forecast data with the actual observations during the forecast period and the simultaneous implementation of alternative models to improve the understanding of the impact of different parameters on the reliability of the used forecast algorithms. We added a corresponding note to the list of already existing realizations of the CITYZER

ecosystem.

The intention of the concept was the possibility to connect a wide range of environmental sensor types, networks and other data sources to the system. These were intentionally not specified further to avoid the impression that alternative approaches would not be possible. Following the reviewers' suggestions we nevertheless added two examples from different fields: a sensor network monitoring water-related effects in Southern Finland using the same interface standard approach, and the access to publicly or commercially available satellite data, which also could be integrated with the CITYZER ecosystem if needed. The utilization of the CITYZER approach as contributor of services to the SmartCity concept is under preparation, and we thank the reviewer for the idea to point it out in this context.

The low technical quality of the figures mentioned by the reviewer is actually due to the publisher's review process. While high resolution figures are provided in separate files for inclusion into the final publication, the discussion version intentionally only contains low quality place holders to indicate contents and location in the finalized version. Found inconsistencies between the figures and the text were corrected. Especially figure 5 was misleading as the process flow was supposed to go from top to bottom. In the updated version this is strictly enforced also for indicated requests from the application side.

Only the application server uses a data base approach to manage user registration and the localization of relevant available data. As it is kept outside the core system any data base approach could be implemented to serve this purpose. In the demonstration version a Linux based public domain MySQL version was used. The authors added this information to the text.

The consistency of the data sources' time stamps is crucial for meaningful combination of a wide range of input data for a data driven forecast model. As the time compatibility in this case has to be only consistent with the time resolution of the forecast model,

the requirements for the correctness of the data time information is only in the order of minutes. Nevertheless the control module ingesting new data into the storage has to ensure to exclude or at least flag data with significantly wrong time stamps. A related note was added to the text.

The way different sensors, sensor groups or complete sensor networks are integrated into a CITYZER-compatible system was intentionally left outside this article except for the standard used for the actual data transfer. Nevertheless the system is designed to cope with the addition or removal of sensors while operational provided that the data from newly added sensors contain sufficiently detailed location definitions.

Following the reviewer's suggestion a new final section with conclusions is added